# Effects of Reversal of Hypotension on Cerebral Microcirculation and Metabolism in Experimental Sepsis

**DOI:** 10.3390/biomedicines10040923

**Published:** 2022-04-18

**Authors:** Fabio Silvio Taccone, Fuhong Su, Xinrong He, Lorenzo Peluso, Katia Donadello, Sabino Scolletta, Daniel De Backer, Jean-Louis Vincent

**Affiliations:** 1Department of Intensive Care, Erasme Hospital, Université Libre de Bruxelles (ULB), 1070 Brussels, Belgium; lorenzopeluso80@gmail.com (L.P.); katia.donadello@univr.it (K.D.); jlvincent@intensive.org (J.-L.V.); 2Laboratoire Experimental des Soins Intensifs, Department of Intensive Care, Hôpital Erasme, Université Libre de Bruxelles (ULB), 1070 Brussels, Belgium; fuhong.su@ulb.be (F.S.); drxinronghe@gmail.com (X.H.); sabino.scolletta@dbm.unisi.it (S.S.); ddebacke@ulb.ac.be (D.D.B.); 3Unit of Anesthesiology and Intensive Care B, AOUI-University Hospital Integrated Trust of Verona, University of Verona, 37129 Verona, Italy; 4Anestesia e Terapia Intensiva, Azienda Ospedaliera Universitaria Senese, 53100 Siena, Italy; 5Intensive Care Department, CHIREC Hospitals, 1160 Brussels, Belgium

**Keywords:** sepsis, cerebral microcirculation, brain oxygenation, brain metabolism, tissue perfusion

## Abstract

The effects of reversal of hypotension on the cerebral microcirculation, oxygenation, and metabolism in septic shock remain unclear. In 12 sheep, peritonitis was induced by injection of feces into the abdominal cavity. At the onset of septic shock (mean arterial pressure (MAP) < 65 mmHg, unresponsive to fluid challenge), a norepinephrine infusion was titrated in eight sheep to restore a MAP ≥ 75 mmHg; the other four sheep were kept hypotensive. The microcirculation of the cerebral cortex was evaluated using side-stream dark-field video-microscopy. Brain partial pressure of oxygen (PbtO_2_) was measured, and cerebral metabolism was assessed using microdialysis. All animals developed septic shock after a median of 15 (14–19) h. When MAP was raised using norepinephrine, the PbtO_2_ increased significantly (from 41 ± 4 to 55 ± 5 mmHg), and the cerebral lactate/pyruvate ratio decreased (from 47 ± 13 to 28 ± 4) compared with values at shock onset. Changes in the microcirculation were unchanged with restoration of MAP and the glutamate increased further (from 17 ± 11 to 23 ± 16 μM), as it did in the untreated animals. In septic shock, the correction of hypotension with vasopressors may improve cerebral oxygenation but does not reverse the alterations in brain microcirculation or cerebral metabolism.

## 1. Introduction

Sepsis still represents a major health issue, with a persistently high mortality rate, despite considerable progress in diagnosis and therapy [1]. Associated organ dysfunction can contribute to prolonged hospital stays and long-term sequelae in survivors [2,3]. Inadequate oxygen supply is one of the main determinants of the development of multiple organ failure [4]; however, even when global hemodynamics are restored, tissue hypoperfusion may persist, in particular due to microcirculatory alterations [5]. Microcirculatory alterations, in particular diffusion and convection abnormalities, can be quantified at the bedside using specific imaging techniques, which may have therapeutic implications [5,6].

Brain dysfunction is common in septic patients and is associated with poor outcome [7]. Although its pathophysiology remains complex and incompletely understood [8], brain hypoperfusion is seen as a key component in its development. In particular, low cerebral blood flow (CBF), impaired CBF autoregulation, and low oxygen saturation have been observed in a large proportion of septic patients and associated with mortality [9,10,11]. Experimental studies have also highlighted that early cerebral microcirculatory abnormalities may occur during sepsis, even before systemic hemodynamics are altered [12]. In septic animals, these microvascular alterations can contribute to the development of electrophysiological abnormalities and neurological dysfunction [13,14]. These microcirculatory abnormalities are also associated with a decrease in cerebral oxygenation and, in the shock state, with impaired brain metabolism [15]; nevertheless, whether such changes are secondary to hypotension or to other septic metabolic processes remains unclear. As, in clinical practice, hypotension is usually corrected by the administration of vasopressors, the aim of this study was to evaluate whether reversal of hypotension using norepinephrine, as recommended in sepsis guidelines, may improve cerebral microcirculation, oxygenation, and metabolism.

## 2. Methods 

The study protocol was approved by the Institutional Review Board for Animal Care of the Free University of Brussels, Brussels, Belgium. Care and handling of the animals followed National Institutes of Health guidelines.

### 2.1. Experimental Animals

Twelve female sheep, weighing between 27 and 35 kg, were fasted for 24 h with free access to water prior to the experiment. On the day of the experiment, the animals were premedicated using intramuscular midazolam (0.25 mg/kg, Dormicum, Roche SA, Anderlecht, Belgium) and ketamine hydrochloride (20 mg/kg, Imalgine, Merial, Lyon, France) and then placed in the supine position. A peripheral venous 18-gauge catheter (Surflo IV Catheter, Terumo Medical Company, Leuven, Belgium) was used to cannulate the cephalic vein and fentanyl citrate (30 µg/kg—Janssen, Beerse, Belgium) and rocuronium bromide (0.1 mg/kg—Esmeron, Organon, Amsterdam, the Netherlands) were administered intravenously for endotracheal intubation (8 mm endotracheal tube, Hi-Contour, Mallinckrodt Medical, Athlone, Ireland). All sheep were thereafter sedated with a continuous intravenous administration of midazolam (0.2 mg/kg*h), ketamine hydrochloride (0.5 mg/kg*h), and morphine (0.2 mg/kg*h), and muscular blockade was given (rocuronium, 10 mcg/kg*h) throughout the experiment to avoid movement artefacts. Boluses of fentanyl (5 mg) were administered if needed in case of tachycardia and/or hypertension suggesting insufficient anesthesia. Mechanical ventilation (Servo 900 C ventilator; Siemens-Elema, Solna, Sweden) was initiated using the following settings: a tidal volume of 10 mL/kg, a respiratory rate of 12–16 breaths/min, a positive end-expiratory pressure (PEEP) of 5 cm H_2_O, an inspired oxygen fraction (FiO_2_) of 1, an inspiratory time to expiratory time ratio of 1:2, and a square-wave pattern. The respiratory rate was adjusted to maintain an end-tidal carbon dioxide pressure (etCO_2_, 47210A Capnometer; Hewlett Packard GmbH, Boehlingen, Germany) between 35 and 45 mmHg before arterial cannulation was performed. A 60 cm plastic tube (inner-diameter 1.8 cm) was inserted into the stomach to drain its content and to prevent rumen distension. A 14F Foley catheter (Beiersdorf AG, Hamburg, Germany) was placed in the bladder to collect the urine output throughout the experiment.

### 2.2. Surgical Procedures

The right femoral artery and vein were surgically exposed. A 6F arterial catheter (Vygon, Cirencester, UK) was introduced into the femoral artery and connected to a pressure transducer (Edwards Lifescience, Irvine, CA, USA) zeroed at mid-chest level. An introducer was inserted through the femoral vein, and a 7F pulmonary artery catheter (Edwards Lifesciences, Irvine, CA, USA) was advanced into the pulmonary artery. A midline laparotomy was then performed. After surgical opening of the cecum, 1.5 g/kg body weight of feces was collected; the cecum was then closed, and the area around the cut was disinfected with iodine solution. An additional suture was inserted to prevent contamination and the cecum was returned to the abdominal cavity. A large plastic tube was inserted through the abdominal wall for later injection of feces. The abdomen was then closed in two layers. During the surgical operation, Ringer’s lactate and 6% hydroxyethyl starch (HES) solutions were infused at rates of 1 mL/kg*h and 2 mL/kg*h, respectively. After abdominal surgery, the animals were turned to the prone position and allowed to stabilize. A bilateral craniotomy was performed in all animals using a high-speed drill and a fine wire saw (Aesculap-Werke AG, Tuttlingen, Germany); two holes were opened and connected until a segment of bone (bone flap) of about 3 × 3 cm was created in the sheep’s frontal bone, as previously described [15]. The dura covering the frontal lobes was then opened in a large incision, carefully avoiding any cortical damage. The left and right frontal lobes were exposed and bleeding from the skull was controlled using surgical wax. At the end of the procedure, the skin flaps were sutured in place. The craniotomy holes were then protected by wet sterile gauzes, avoiding any contact with the brain cortex. Brain desiccation was prevented by local administration of 2 mL of NaCl 0.9% saline solution hourly.

### 2.3. Monitoring and Measurements

Mechanical ventilation parameters were adjusted to keep PaO_2_ between 80 and 120 mmHg and PaCO_2_ between 35 and 45 mmHg, according to repeated hourly blood gas analysis (ABL500; Radiometer, Copenhagen, Denmark). Hemoglobin concentration and oxygen saturation were measured with an analyzer calibrated for ruminant animals (OSM3; Radiometer). Peak airway pressure, plateau airway pressure, expiratory gas flow, and FiO_2_ were recorded hourly. The total amount of blood withdrawn for gas analyses was estimated at around 60 mL (i.e., around 3% of each sheep’s estimated total blood volume). All monitored variables were recorded every 60 min. Measurements of mean arterial pressure (MAP), pulmonary arterial pressure, right atrial pressure, and pulmonary artery occlusion pressure (PAOP) were obtained at end-expiration (Sirecust 404; Siemens, Erlangen, Germany). Core temperature and cardiac output (Vigilance; Baxter, Edwards Critical Care) were monitored continuously. Body surface area, cardiac index, stroke volume index, and systemic vascular resistance (SVR) were calculated using standard formulas.

### 2.4. Cerebral Microcirculation

The microvascular network of the cerebral cortex was visualized on the left lobe using an SDF video-microscopy system (MicroScanTM, MicroVisionMedical Inc., Amsterdam, The Netherlands) with a 5× imaging objective giving 326× magnification. The lens of the imaging device was covered with a disposable sterile cap and was applied without pressure to the cerebral frontal cortex. Because of brain pulsatility, this was best accomplished by placing the device on a metallic arm for stabilization (Giesseci, Avellino, Italy). Lack of pressure was ensured by preservation of flow in large vessels [16]. At least five videos from different areas, each of a minimum duration of 20 s, were recorded on an external hard-disk using a computer and a video card (MicroVideo; Pinnacle Systems, Mountain View, CA, USA). The images were then stored under a random number for further analysis. An investigator blinded to group allocation and time later analyzed these sequences semi-quantitatively [16]. The vascular density was calculated as the number of vessels crossing these lines divided by the total length of the lines. The type of flow was defined as continuous, intermittent, or absent and the Mean Flow Index (MFI) was computed [16]. Vessel size was determined using a micrometer scale and the vessels were separated into large and small vessels using a diameter cutoff value of 20 μm [15]. Small vessel perfusion was defined as the proportion of small perfused vessels (PPV) and calculated as the number of capillaries continuously perfused during the 20 s observation period divided by the total number of vessels of the same type. Functional capillary density (FCD) was calculated as the product of capillary density and perfused vessel density of vessels of the same type. The Heterogeneity Index for PPV (PPV-HI) was also calculated [15]. In each animal, the data from the investigated areas were averaged for each time point.

### 2.5. Cerebral Oxygenation and Metabolism

An intracranial brain oxygen pressure (PbtO_2_) catheter was introduced through the right cranial hole and connected to a specific monitor (Brain Tissue Oxygen Monitoring, AC31, Integra Lifesciences, Zaventem, Belgium), in which the temperature was manually adjusted to blood temperature. Probe function was confirmed by an oxygen challenge (FiO_2_ 1.0 for 5 min). The CMA 20 catheter was introduced close to the PbtO_2_ catheter and perfused by a miniaturized infusion pump (CMA 107, CMA Microdialysis AB, Solna, Sweden) with a CNS perfusion fluid (148 mM NaCl, 2.7 mM KCl, 1.2 mM CaCl_2_, and 0.85 mM MgCl_2_; osmolality 305 mOsm/kg; pH 6.0) at a flow rate of 1.0 μL/min. This flow rate guarantees an almost 50% recovery rate for molecules of less than 20 kDa and was selected to provide additional fluid for further research on brain metabolites. To allow for probe equilibration, data from the first hour after placement were not used. After one hour of stabilization, the perfusate was collected every 60 min in specific microvials. Samples were analyzed for lactate, pyruvate, glycerol, glutamate, and glucose using an automatic analyzer (CMA 600 Microdialysis Analyzer, CMA Microdialysis AB, Stockholm, Sweden). The lactate/pyruvate ratio (LPR) was automatically calculated.

### 2.6. Experimental Protocol

After the surgical procedures, baseline measurements, including cerebral microcirculation, oxygenation, and metabolism, were obtained. Feces were then injected into the abdominal cavity (sepsis group). Ringer’s lactate solution (RL) and 6% hydroxyethyl starch solution (HES, Voluven, Fresenius Kabi, Schelle, Belgium) were titrated to prevent hypovolemia, i.e., to maintain the PAOP at baseline levels, as previously reported [15]. Moreover, in case of a reduction in MAP of more than 15% from baseline, and/or lactate > 2.0 mmol/L, and/or SvO_2_ < 65%, or a hemoglobin concentration increase of more than 1.5 g/dL, a bolus of 20 mL/Kg of RL was also considered. At the onset of septic shock (mean arterial pressure (MAP) < 65 mmHg, unresponsive to fluid challenge), norepinephrine was administered in 8 sheep and titrated to restore a MAP ≥ 75 mmHg (NE group); the 4 other sheep were kept hypotensive (No-NE group). Measurements of all hemodynamic and cerebral variables were obtained at baseline, 6 h and 12 h, at shock onset, and 2 h after the start of norepinephrine infusion.

### 2.7. Statistical Analysis

Statistical analysis was performed using SPSS 24.0 for Windows (SPSS, Chicago, IL, USA). Data are presented as mean ± SD or median (ranges), as appropriate. The significance of differences in the measured variables between the different time-points was analyzed using a mixed-effects model for repeated measures, followed by a Sidak post hoc correction. A *p* value < 0.05 was considered statistically significant.

## 3. Results

All animals developed septic shock after a median time of 15 (14–19) h after feces injection. The main hemodynamic, respiratory, and metabolic variables are shown in Table 1 and Figure 1.

MAP was higher and lactate levels lower after 2 h of norepinephrine administration in the NE group compared with the untreated animals. All other variables were similar between the two groups at the different time-points.

Cerebral PPV, FCD, and MFI decreased and PPV-HI increased significantly until shock onset in all animals (Figure 2). Concomitantly, the PbtO_2_ decreased significantly, and brain metabolism deteriorated; in particular, the LPR increased, as did glutamate and glycerol (Figure 3). The norepinephrine infusion (median dose 2.8 (range: 1.8–3.2) mcg/kg × min) increased MAP (to 79 (range: 74–82) mmHg)—Figure 1); in untreated animals, MAP remained stable.

In the NE group, the PbtO_2_ increased significantly compared with the No-NE group (Figure 3), but did not reach the same values as at baseline. In the NE group, the LPR decreased but remained above the threshold of 25 for most of measurements; in the No-NE group, the LPR continued to increase (Figure 3). Microcirculatory alterations remained unchanged after norepinephrine administration, with similar PPV, FCD, MFI, and PPV-HI values in the two groups (Figure 3). Glutamate and glycerol brain levels increased in both groups.

## 4. Discussion

In this experimental model of ovine sepsis, the key findings are: (a) alterations in cortical cerebral microcirculation cannot be reversed by the administration of norepinephrine; (b) the reduction in brain oxygenation and the increase in LPR, as markers of tissue hypoxia and anaerobic metabolism, respectively, are partially reversed by the correction of hypotension, although they remain below baseline and physiological values; (c) persisting increases in cerebral glycerol or glutamate, suggesting ongoing excitotoxicity and cellular damage, are unresponsive to the reversal of hypotension using norepinephrine.

In this study, the septic animals had already undergone initial fluid resuscitation and had been in shock for two hours; no other changes in treatment were made after vasopressor initiation to ensure that alterations in the measured variables were not influenced by factors other than the increase in MAP. Norepinephrine is presently considered as the vasopressor of choice in septic shock [17]; moreover, even in patients with acute traumatic brain injury, the response of cerebral perfusion pressure and oxygen delivery is more predictable with norepinephrine than with dopamine and norepinephrine is currently used to improve brain perfusion in this setting [18]. Whether non-adrenergic agents [19] might provide different effects on the cerebral microcirculation, oxygenation, and metabolism during sepsis needs to be further evaluated.

How can one explain the lack of an effect of effective vasopressor support on the cerebral microcirculation? As tissue perfusion and microvascular flow depend, among other factors, on the diameter of the vessel and the applied pressure gradient, the vasoconstricting effects of norepinephrine may increase the cerebrovascular resistance (i.e., constriction in pre-capillary arteries) and this may limit the increase in microvascular flow induced by the increase in perfusion pressure. Another potential explanation is that, if cerebral blood flow autoregulation was preserved, a regional microvascular response may have occurred to maintain constant brain perfusion over time, thus resulting in no change in the vascular tone or in the microcirculatory flow [20]. However, norepinephrine was initiated when the MAP was less than 65 mmHg, i.e., below the commonly accepted inferior threshold of CBF autoregulation; as such, a significant change in the cerebrovascular resistance should have been observed after the MAP increase. Finally, it is possible that microcirculatory alterations are not only dependent on systemic hemodynamics but also on the persistence of the septic process [21]. In one experimental study in septic rats, norepinephrine administration improved the evoked potential amplitudes [22], probably because of an enhanced activation of cortical neuronal fields, but did not improve the microvascular flow.

Brain hypoxia has already been observed in experimental sepsis [15]. In another study conducted in conscious sheep, sepsis induced by the injection of live *Escherichia coli* for 24 h was associated with a progressive decrease in MAP and PaO_2_ over time, which was associated with a significant reduction in PbtO_2_ (from 32.2 ± 10.1 to 18.8 ± 11.7 mmHg after 3 h and to 22.8 ± 5.2 mmHg after 24 h of sepsis) [23]. However, the reduction in PbtO_2_ observed in this study might have been secondary to hypotension (associated with a decrease in cerebral blood flow) and hypoxemia (with reduced arterial oxygen content), which can be easily corrected in clinical practice. In our experimental model, as PbtO_2_ changes could not be entirely restored to baseline values but remained significantly higher in animals treated with norepinephrine compared with others, we could argue that the reduction in cerebral oxygenation during sepsis could be initially related to early microvascular alterations and further aggravated by systemic hemodynamic factors, such as hypotension; although hypotension and related tissue hypoxia could be easily reversed by the administration of vasopressors, persistent microvascular alterations contributed to maintain lower cerebral oxygenation values than initial values. Importantly, cerebral microcirculation was assessed at the pial level, while the PbtO_2_ catheter generally assesses oxygenation within the gray matter; regional blood flow to these areas may be dissociated in shock, at least in ischemia/reperfusion models [24]; no data in sepsis are available.

The switch toward anaerobic metabolism, as indicated by an increased LPR, was also partially, but not entirely, reversed by norepinephrine. Cerebral metabolic derangements are poorly characterized during sepsis. No changes in brain metabolism over time were observed in another swine model of resuscitated sepsis [25]; however, the animals developed a hyperdynamic state with hyperlactatemia but not hypotension. In another model of porcine endotoxemic sepsis, cerebral levels of glutamate, glycerol, and LPR were elevated [26]. Increased cerebral LPR values were also observed in another endotoxic model in pigs [27]; however, endotoxin infusion does not adequately mimic the septic shock observed in clinical practice. As such, anaerobic cerebral metabolism may occur as a combination of systemic and microvascular impairment and can be partially corrected by the administration of vasopressors. However, alterations in neuro-energetics with exacerbated activity of the glutamate/glutamine cycle and increased cellular distress are not sensitive to hemodynamic intervention, which may explain the persistently high levels of glycerol and glutamate in our model. In our model, the correction of hypotension was not sufficient to reverse cellular metabolism disturbances; the persistence of microvascular abnormalities could be one potential explanation, with the LPR remaining above normal values and the substrate supply being limited by altered regional flow, thus resulting in cellular damage. Whether other interventions that may reduce the LPR increase during sepsis, such as hyperoxia [28], could also alter this neuro-energetic dysfunction requires further analysis. Other therapeutic options, such as administration of ketone bodies or hypertonic lactate [29,30], which have a significant effect on cerebral glycolysis and metabolic distress, might have a potential neuroprotective effect in this setting. Finally, norepinephrine can also stimulate neuronal metabolic activity (i.e., increase glycolysis, thus raising cerebral lactate and pyruvate levels) and alter glutamate buffering [31] (i.e., increase interstitial glutamate levels), which may have influenced our findings.

There are some limitations to this study. First, we visualized only pial vessels and the frontal cortex, and these areas may not be representative of deeper brain structures, including white matter and the brain stem. Second, we did not measure regional CBF, flow autoregulation, or cellular function, and our findings cannot be correlated with cerebral perfusion or neuronal electrical activity. Third, we did not perform any post mortem histological examination of brain parenchyma. Fourth, our model is lethal and we could not evaluate any association between microvascular disturbances and clinical neurological abnormalities. Fifth, we did not study different levels of MAP and their relative effects on cerebral microvascular perfusion. Sixth, source control and antibiotics were not provided as suggested to increase the translatability of research findings on animal models of sepsis [32]; however, we aimed to reproduce the different phases of sepsis (i.e., hyperdynamic; organ dysfunction; shock) within the 18 h following fecal injection. As such, this study should be considered more as a mechanistic research investigation than an evaluation of therapeutic effectiveness. Seventh, we evaluated sepsis in sheep (while other species could also be used) using only female animals and a model of fecal peritonitis (instead of lipopolysaccharide injection or intraperitoneal or intratracheal bacterial injection). Although this might be considered a limitation, no individual model is a perfect vehicle for drug therapy testing or studying physiological mechanisms [33]. Finally, these data are difficult to extrapolate to the human setting; the brain microcirculation is still impossible to monitor and visualize in clinical practice without direct exposure of the cerebral cortex after craniectomy, so few data on microvascular abnormalities are available and those that do exist are only for primary brain injury.

## 5. Conclusions

In this ovine model of sepsis, reversal of hypotension using norepinephrine resulted in partial correction of tissue hypoxia and did not influence anaerobic metabolism. Persistent alterations of the cerebral microcirculation may explain these findings.

## Figures and Tables

**Figure 1 biomedicines-10-00923-f001:**
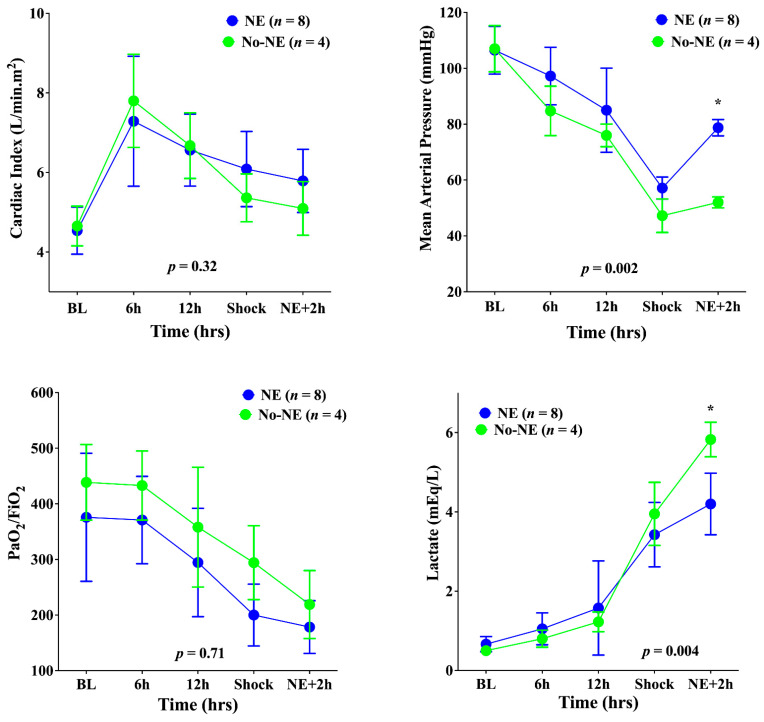
Cardiac output, mean arterial pressure, the PaO_2_/FiO_2_ ratio, and lactate levels in the two groups at different study time-points. NE, norepinephrine. Data are presented as mean ± SEM. * *p* < 0.05 from Sidak post hoc analysis.

**Figure 2 biomedicines-10-00923-f002:**
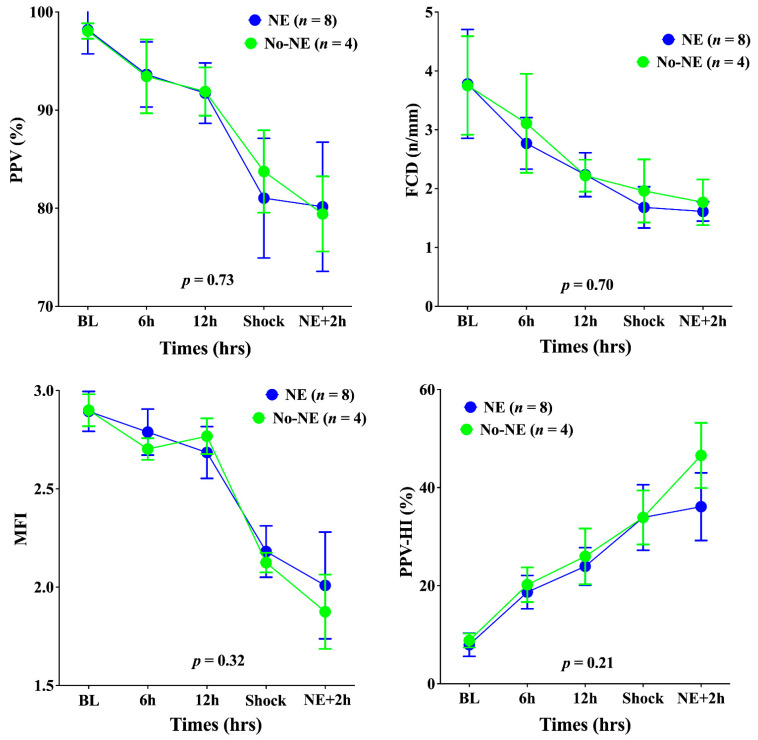
Time-course of the proportion of small perfused vessels (PPV), the functional capacity density (FCD), the mean flow index (MFI), and the heterogeneity index of PPV (PPV-HI) in the cerebral microcirculation in the two groups of animals. NE, norepinephrine. Data are presented as mean ± SEM.

**Figure 3 biomedicines-10-00923-f003:**
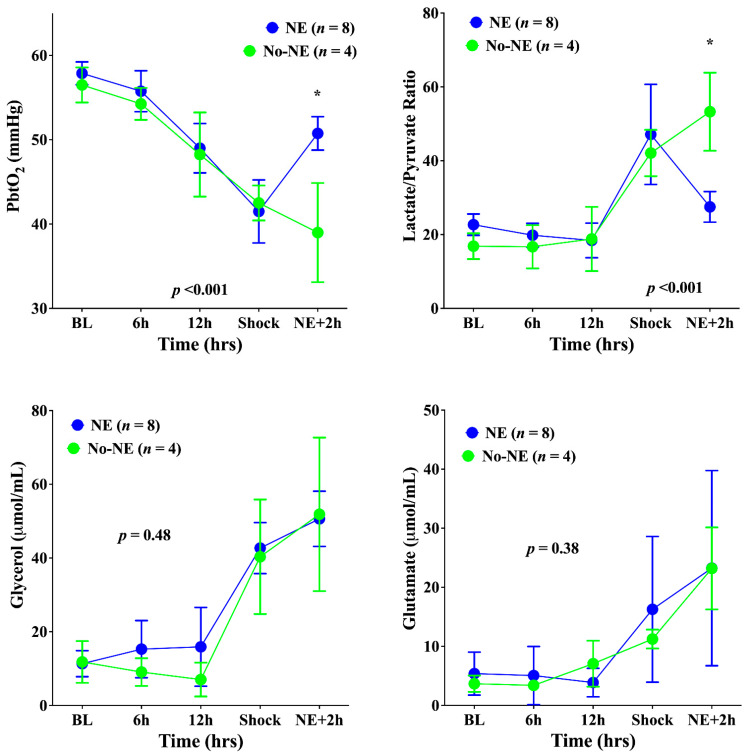
Time-course of the brain oxygen pressure (PbtO_2_), the cerebral lactate/pyruvate ratio, cerebral glycerol, and cerebral glutamate in the two groups. NE, norepinephrine. Data are presented as mean ± SEM. * *p* < 0.05 from Sidak post hoc analysis.

**Table 1 biomedicines-10-00923-t001:** Hemodynamic and respiratory variables in the two groups at different study time-points. The *p* value corresponds to the mixed-effects model analysis. HR, heart rate; MPAP, mean pulmonary artery pressure; PAOP, pulmonary artery occlusive pressure; TPC, thoraco-pulmonary compliance; NE, norepinephrine. Data are given as mean ± SD.

		Baseline	6 h	12 h	Shock	NE + 2 h	*p* Value
**Temperature, °C**	NE	39.6 ± 0.5	40.1 ± 0.4	40.6 ± 0.6	41.3 ± 0.6	41.1 ± 0.6	0.75
	No-NE	39.7 ± 0.3	40.2 ± 0.2	40.8 ± 0.6	41.8 ± 0.2	41.3 ± 0.7	
**HR, beats/min**	NE	113 ± 14	144 ± 21	147 ± 19	145 ± 25	151 ± 21	0.47
	No-NE	120 ± 8	156 ± 29	154 ± 19	141 ± 6	140 ± 11	
**MPAP, mmHg**	NE	15 ± 2	13 ± 3	16 ± 3	18 ± 5	17 ± 6	0.39
	No-NE	14 ± 3	15 ± 2	15 ± 6	20 ± 4	18 ± 4	
**PAOP, mmHg**	NE	4 ± 1	3 ± 2	4 ± 1	5 ± 2	5 ± 1	0.82
	No-NE	4 ± 1	3 ± 1	4 ± 1	5 ± 2	5 ± 1	
**PaCO_2_, mmHg**	NE	39 ± 3	38 ± 1	36 ± 3	42 ± 7	43 ± 8	0.66
	No-NE	41 ± 5	36 ± 3	37 ± 1	40 ± 8	41 ± 11	
**PaO_2_, mmHg**	NE	133 ± 9	129 ±9	119 ± 14	110 ± 6	106 ± 11	0.11
	No-NE	117 ± 14	117 ± 5	112 ± 18	111 ± 4	109 ± 9	
**pH**	NE	7.41 ± 0.04	7.38 ± 0.03	7.36 ± 0.03	7.25 ± 0.08	7.26 ± 0.08	0.52
	No-NE	7.37 ± 0.06	7.37 ± 0.06	7.36 ± 0.04	7.25 ± 0.10	7.28 ± 0.13	
**TPC, mL/mmHg**	NE	19 ± 4	17 ± 2	16 ± 4	12 ± 2	14 ± 2	0.69
	No-NE	21 ± 4	17 ± 3	16 ± 5	13 ± 3	15 ± 5	
**Hemoglobin, g/dL**	NE	9.8 ± 1.4	11.6 ± 1.1	12.5 ± 1.0	13.2 ± 1.8	12.7 ± 2.0	0.17
	No-NE	10.5 ± 0.7	11.2 ± 1.2	11.1 ± 1.2	11.9 ± 2.7	12.1 ± 2.9	
**Urine Output, mL**	NE	89 ± 66	319 ± 98	689 ± 89	798 ± 129	804 ± 128	0.49
	No-NE	105 ± 98	389 ± 112	703 ± 109	811 ± 133	813 ± 135	
**Fluid Amount, mL**	NE	317 ± 99	1423 ± 616	1840 ± 622	2310 ± 777	2490 ± 971	0.61
	No-NE	299 ± 111	1530 ± 866	1980 ± 855	2198 ± 1012	2318 ± 999	

## Data Availability

Data regarding this study are available on request to the corresponding author.

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
