# Peer review of "Effects of Reversal of Hypotension on Cerebral Microcirculation and Metabolism in Experimental Sepsis"

_biomedicines, 2022, doi:10.3390/biomedicines10040923_

Round 1
Reviewer 1 Report
Thank you for your very interesting and though stimulating submission.
The presented study focuses on various aspects of cerebral perfusion changes in an ovine model of peritoneal septic shock. The study set up was very sophisticated with measurements of cerebral microcirculation, oxygenation, and metabolism. The authors are able to show that despite restoration of cerebral pressures with vasopressor application and improvements in mean arterial pressures (suspected globally) and subsequent improvements in oxygenation of the cerebellar tissues continued anaerobic metabolism persists.
SOURCE CONTROL/ Antibiotics
In the setting of peritoneal sepsis ground principles of sepsis care apply including source control. [ Jimenez MF, Marshall JC; International Sepsis Forum. Source control in the management of sepsis. Intensive Care Med. 2001;27 Suppl 1:S49-62. doi: 10.1007/pl00003797. PMID: 11307370.] The placement of a fecal matter into the abdominal cavity was needed for the introduction of bacteria into a sterile cavity and poses a severe immunologic insult. Without removal of this material along with antibiotic treatment this immunologic insult is ongoing and may contribute to various irreversible metabolic changes in regard to the brain metabolism and possibly ongoing immune alterations. Riche’ showed in an observational study of patients with peritonitis that the patients with the most severe condition (shock, non-survivors) an array of pro and anti-inflammatory cytokines in different concentrations in the abdominal compartment and the circulation. [Riché F, Gayat E, Collet C, et al. Local and systemic innate immune response to secondary human peritonitis. Crit Care. 2013;17(5):R201. Published 2013 Sep 12. doi:10.1186/cc12895]. With the changes in cytokines in te blood it could be assumed that distant metabolic and possibly transcriptomic changes are initiated and will be ongoing without sufficient source control contributing to the observed alterations in brain metabolism despite the restoration of adequate perfusion.
- Could you please describe your approach to source control in the study.
- If you wanted to assess the natural occurring changes of continued septic insult you should state this.
Resuscitation
Part of resuscitation from shock to restore adequate perfusion vasopressors AND FLUIDS are used. Dr De Backer is a co-author on an article regarding “FLUID RESUSCIATION and VASOPRESSOR THERAPY” published last year. [Lat, Ishaq PharmD, FCCM1; Coopersmith, Craig M. MD, MCCM2; De Backer, Daniel MD, PhD3; for the Research Committee of the Surviving Sepsis Campaign Members of the Surviving Sepsis Campaign Research Committee contributing to this article are as follows Co-chair, Atlanta, GA Co-chair, Brussels, Belgium Manhasset, NY Consultant, Seattle, WA Barcelona, Spain San Francisco, CA Pune, India Utrecht, the Netherlands Chicago, IL Sao Paulo, Brazil Atlanta, GA Dublin, Ireland New York, NY Consultant, London, United Kingdom. The Surviving Sepsis Campaign: Fluid Resuscitation and Vasopressor Therapy Research Priorities in Adult Patients, Critical Care Medicine: April 2021 - Volume 49 - Issue 4 - p 623-635 doi: 10.1097/CCM.0000000000004864] In paragraph 2.6 Experimental Protocol line 170 to 175 you describe administration of RL and 6% HES to maintain PAOP. Could you please describe the fluid resuscitation in more detail. (Supplemental materials may be helpful to keep the text focused).
à What was the approach to fluid administration and titration?
- Where you able to measure other hemodynamic markers indicating adequate volume:
CVP, PPV, CO, CVI etc. (Could you please provide more detail).
- Was the fluid amount fixed for all animals based on body weight?
- Did you titrate fluids to lactate clearance parameters for the resuscitation/ VP group?
Cerebral Pressures/ Intracranial pressure
Dr Taccone has published several very important articles looking at various aspects of cerebral autoregulation and the development of sepsis associated encephalopathy in the ovine model of sepsis described in this study. [Ferlini, L., Su, F., Creteur, J. et al. Cerebral autoregulation and neurovascular coupling are progressively impaired during septic shock: an experimental study. ICMx 8, 44 (2020). https://doi.org/10.1186/s40635-020-00332-0].
- Did the authors measure changes in intracranial pressures (ICP) at different time points.
A comprehensive review by Dr Sonneville et al [Sonneville R, Verdonk F, Rauturier C, et al. Understanding brain dysfunction in sepsis. Ann Intensive Care. 2013;3(1):15. Published 2013 May 29. doi:10.1186/2110-5820-3-15] pointed to MRI changes seen in various studies such as cytotoxic edema and vasogenic edema. These changes in conjunction with possible ongoing inflammatory stimuli could be part of the observed continued changes in regard to glutamate metabolism despite restoration of the cerebral perfusion.
Other minor questions
- Could you please briefly state why only female sheep were used for the experimental design.
- Figure 4 (Page 8, line 228) is missing the legend.
- The legends for Figure 1-3 should be underneath the figures, but there seems to be some formatting mishap.
Thank you again for your submission
Author Response
Reviewer 1
- The presented study focuses on various aspects of cerebral perfusion changes in an ovine model of peritoneal septic shock. The study set up was very sophisticated with measurements of cerebral microcirculation, oxygenation, and metabolism. The authors are able to show that despite restoration of cerebral pressures with vasopressor application and improvements in mean arterial pressures (suspected globally) and subsequent improvements in oxygenation of the cerebellar tissues continued anaerobic metabolism persists.
Authors’ response: we thank the reviewer for the nice comment and summary.
- In the setting of peritoneal sepsis ground principles of sepsis care apply including source control [Jimenez MF, Marshall JC; International Sepsis Forum. Source control in the management of sepsis. Intensive Care Med. 2001;27 Suppl 1:S49-62. doi: 10.1007/pl00003797. PMID: 11307370.]The placement of a fecal matter into the abdominal cavity was needed for the introduction of bacteria into a sterile cavity and poses a severe immunologic insult. Without removal of this material along with antibiotic treatment this immunologic insult is ongoing and may contribute to various irreversible metabolic changes in regard to the brain metabolism and possibly ongoing immune alterations. Riche’ showed in an observational study of patients with peritonitis that the patients with the most severe condition (shock, non-survivors) an array of pro and anti-inflammatory cytokines in different concentrations in the abdominal compartment and the circulation. [Riché F, Gayat E, Collet C, et al. Local and systemic innate immune response to secondary human peritonitis. Crit Care. 2013;17(5):R201. Published 2013 Sep 12. doi:10.1186/cc12895]. With the changes in cytokines in the blood it could be assumed that distant metabolic and possibly transcriptomic changes are initiated and will be ongoing without sufficient source control contributing to the observed alterations in brain metabolism despite the restoration of adequate perfusion. Could you please describe your approach to source control in the study. If you wanted to assess the natural occurring changes of continued septic insult you should state this.
Authors’ response: The reviewer is right – we did not specifically control the source of infection, in order to reproduce the different phases of sepsis (hyperdynamic; organ dysfunction; shock) within the 15-18 hours following fecal injection. This is a main limitation of the study and has been highlighted as such.
- Part of resuscitation from shock to restore adequate perfusion vasopressors AND FLUIDS are used. Dr De Backer is a co-author on an article regarding “FLUID RESUSCIATION and VASOPRESSOR THERAPY” published last year. [Lat, Ishaq PharmD, FCCM1; Coopersmith, Craig M. MD, MCCM2; De Backer, Daniel MD, PhD3; for the Research Committee of the Surviving Sepsis Campaign Members of the Surviving Sepsis Campaign Research Committee contributing to this article are as follows Co-chair, Atlanta, GA Co-chair, Brussels, Belgium Manhasset, NY Consultant, Seattle, WA Barcelona, Spain San Francisco, CA Pune, India Utrecht, the Netherlands Chicago, IL Sao Paulo, Brazil Atlanta, GA Dublin, Ireland New York, NY Consultant, London, United Kingdom. The Surviving Sepsis Campaign: Fluid Resuscitation and Vasopressor Therapy Research Priorities in Adult Patients, Critical Care Medicine: April 2021 - Volume 49 - Issue 4 - p 623-635 doi: 10.1097/CCM.0000000000004864]In paragraph 2.6 Experimental Protocol line 170 to 175 you describe administration of RL and 6% HES to maintain PAOP. Could you please describe the fluid resuscitation in more detail. (Supplemental materials may be helpful to keep the text focused). What was the approach to fluid administration and titration? Where you able to measure other hemodynamic markers indicating adequate volume: CVP, PPV, CO, CVI etc. (Could you please provide more detail). Was the fluid amount fixed for all animals based on body weight? Did you titrate fluids to lactate clearance parameters for the resuscitation/ VP group?
Authors’ response: We thank the reviewer for the relevant question. We have published several articles using the same model, which included the administration of 6% HES together with crystalloid solutions (PMID: 29921225, PMID: 28930916, PMID: 28708664, PMID: 28632534). Although HES are not recommended anymore in clinical practice, adding a colloid solution to crystalloids would reduce the total amount of given fluids and potentially reduce the risk of lung overload and intrabdominal hypertension. Fluid administration was titrated based on some parameters, which have been described into the text, accordingly.
- Dr Taccone has published several very important articles looking at various aspects of cerebral autoregulation and the development of sepsis associated encephalopathy in the ovine model of sepsis described in this study. [Ferlini, L., Su, F., Creteur, J. et al. Cerebral autoregulation and neurovascular coupling are progressively impaired during septic shock: an experimental study. ICMx 8, 44 (2020). https://doi.org/10.1186/s40635-020-00332-0]. Did the authors measure changes in intracranial pressures (ICP) at different time points. A comprehensive review by Dr Sonneville et al [Sonneville R, Verdonk F, Rauturier C, et al. Understanding brain dysfunction in sepsis. Ann Intensive Care. 2013;3(1):15. Published 2013 May 29. doi:10.1186/2110-5820-3-15] pointed to MRI changes seen in various studies such as cytotoxic edema and vasogenic edema. These changes in conjunction with possible ongoing inflammatory stimuli could be part of the observed continued changes in regard to glutamate metabolism despite restoration of the cerebral perfusion.
Authors’ response: We thank the reviewer for this important comment. We did not measure ICP because of the large craniectomy used to expose microcirculation, which would dramatically reduce ICP values. We have cited the reference about glutamate and cerebral edema, as requested.
- Could you please briefly state why only female sheep were used for the experimental design.
Authors’ response: At the moment of study experiments, our laboratory had used only female sheep for years.
- Figure 4 (Page 8, line 228) is missing the legend.
Authors’ response: There was a typo in the manuscript (“4” was for the Discussion section). This has been corrected, accordingly.
- The legends for Figure 1-3 should be underneath the figures, but there seems to be some formatting mishap.
Authors’ response: This has been corrected, as requested.
Reviewer 2 Report
In this original article entitled “Effects of reversal of hypotension on cerebral microcirculation and metabolism in experimental sepsis” Taccone et al. aimed to investigate the effects of reversal of hypotension on cerebral microcirculation, oxygenation and metabolism under septic conditions in vivo. Importantly, sepsis is a life-threating condition (one of the leading causes of death worldwide) with dysregulated systemic host response to microbial pathogens, while the septic shock is a subset of sepsis with circulatory, metabolic and cellular abnormalities. Consequently, multiple organ failure can develop rapidly, resulting in early death. Beside biomarker research, investigation of microcirculatory alterations upon septic shock is also a ‘hot-topic’, which may have therapeutic aspects.
The authors demonstrated that MAP and PbtO2 were significantly increased while lactate concentration and cerebral lactate/pyruvate ratio were decreased in norepinephrine treated animals vs. hypotensive controls, which resulted in partial correction of tissue hypoxia. For this purpose, an experimental animal (sheep) model was used in which sepsis was induced with abdominal injection of feces collected by cecal puncture.
Overall, the manuscript has merits and fits to the scope of the Journal. Despite of these interesting findings, some improvements should be addressed before final decision.
Major comments:
- In this study the Authors used sheep as an animal model for sepsis, however, several other animal models (e.g. dog, pig, etc.) are widely used. What was the reason of choosing the ovine model?
- Based on the literature, numerous methods (endotoxin application, intravascular administration of pathogens or cecal ligation and puncture) are available to induce sepsis or septic shock. What is the state-of-the-art method in this field? Why feces injection was chosen to be used in this study? Insert these details (Q1 and Q2) into the Introduction part of the manuscript indicating relevant references.
- Were control animals (without injection of feces) used to check the efficiency of inducing septic shock? What were the criteria to set up the diagnose of septic shock?
- In the manuscript the reviewer found hemodynamic and respiratory results. Is there any data of blood parameters of inflammation (e.g. CRP, IL-6, PCT, etc.)?
- Only 4 sheep were kept under hypotensive circumstance. For evaluation of the normality of data using the Kolmogorov-Smirnov test, n=4 is too small. In this case, to compare the data of two groups (2 hours after NE vs. No-NE group, unpaired t-test is the recommended test.
- The Results part is too short and incomplete. More details and explanation are required in the text.
- Has the effect of norepinephrine investigated in human sepsis? It is recommended to cite more human-related articles in the Discussion.
Minor comments:
- Figure and Table legends should be completed with more experimental and statistical details.
- On Figures 1-3 mean ± SD values was depicted, however, mean ± SEM would be better.
- The abbreviation of NE-2h timepoint on Figures is somewhat confusing, it would be better to use another label (e.g. after 2h of NE administration, NE+2h).
- Please include the permit number of the ethical approval.
- A list of abbreviations would be helpful for the readers.
Author Response
Reviewer 2
- In this original article entitled “Effects of reversal of hypotension on cerebral microcirculation and metabolism in experimental sepsis” Taccone et al. aimed to investigate the effects of reversal of hypotension on cerebral microcirculation, oxygenation and metabolism under septic conditions in vivo. Importantly, sepsis is a life-threating condition (one of the leading causes of death worldwide) with dysregulated systemic host response to microbial pathogens, while the septic shock is a subset of sepsis with circulatory, metabolic and cellular abnormalities. Consequently, multiple organ failure can develop rapidly, resulting in early death. Beside biomarker research, investigation of microcirculatory alterations upon septic shock is also a ‘hot-topic’, which may have therapeutic aspects. The authors demonstrated that MAP and PbtO2 were significantly increased while lactate concentration and cerebral lactate/pyruvate ratio were decreased in norepinephrine treated animals vs. hypotensive controls, which resulted in partial correction of tissue hypoxia. For this purpose, an experimental animal (sheep) model was used in which sepsis was induced with abdominal injection of feces collected by cecal puncture. Overall, the manuscript has merits and fits to the scope of the Journal. Despite of these interesting findings, some improvements should be addressed before final decision.
Authors’ response: We thank the reviewer for the nice summary of our data.
- In this study the Authors used sheep as an animal model for sepsis, however, several other animal models (e.g. dog, pig, etc.) are widely used. What was the reason of choosing the ovine model?
Authors’ response: This is a very good point raised by the reviewer. As replied also to the reviewer number 1, our group has published more than 30 original articles using this model. Dog research is almost impossible nowadays, also in Belgium, and in particular for lethal models. Pig models have some advantages (i.e. circulatory system very close to humans; large experience in different experimental models; many assays available for laboratory investigations) but also disadvantages (i.e. less hyperdynamic response, less lactate production, more sedatives required) when compared to sheep. Although we agree that some experiments aiming at the evaluation of new therapies in sepsis should be investigated in different species, our study was based on the quantification of a physiological response to increases MAP and it is more mechanistic/physiological (so, acceptable if tested in one species).
- Based on the literature, numerous methods (endotoxin application, intravascular administration of pathogens or cecal ligation and puncture) are available to induce sepsis or septic shock. What is the state-of-the-art method in this field? Why feces injection was chosen to be used in this study? Insert these details (Q1 and Q2) into the Introduction part of the manuscript indicating relevant references.
Authors’ response: We thank the reviewer for this question. The role of different method to induce sepsis in animals has been largely debated in different articles (PMID: 24022070; PMID: 32648582; PMID: 25565638). Many species, including mice and baboons, are remarkably resistant to the toxic effects of bacterial lipopolysaccharide. In general, no individual model is a perfect vehicle for drug therapy testing and complementary model use is optimal for studies on therapy mechanisms. We have reported this more as a Limitation of the study.
- Were control animals (without injection of feces) used to check the efficiency of inducing septic shock? What were the criteria to set up the diagnose of septic shock? In the manuscript the reviewer found hemodynamic and respiratory results. Is there any data of blood parameters of inflammation (e.g. CRP, IL-6, PCT, etc.)?
Authors’ response: As stated before, septic shock reproducibility has been largely presented and documented in several articles from our group, so that a control group was not necessary. In particular for this experiment, we have previously evaluated cerebral microcirculation using sham animals (PMID: 20667108), which documented hemodynamic and cerebral differences between septic and sham animals for most of studied variables.
- Only 4 sheep were kept under hypotensive circumstance. For evaluation of the normality of data using the Kolmogorov-Smirnov test, n=4 is too small. In this case, to compare the data of two groups (2 hours after NE vs. No-NE group, unpaired t-test is the recommended test.
Authors’ response: The reviewer is correct. We have repeated all analyses which gave similar results.
- The Results part is too short and incomplete. More details and explanation are required in the text.
Authors’ response: We kindly disagree with the reviewer. We have presented the data related to the research question. All additional data related to the model has been reported in Table 1 and Supplemental Table 1, accordingly.
- Has the effect of norepinephrine investigated in human sepsis? It is recommended to cite more human-related articles in the Discussion.
Authors’ response: We thank the reviewer for this question. Unfortunately cerebral microcirculation is not evaluable in the human setting (at least not in septic patients) so there are no human data to discuss.
- Figure and Table legends should be completed with more experimental and statistical details. On Figures 1-3 mean ± SD values was depicted, however, mean ± SEM would be better. The abbreviation of NE-2h timepoint on Figures is somewhat confusing, it would be better to use another label (e.g. after 2h of NE administration, NE+2h). Please include the permit number of the ethical approval.
Authors’ response: We have added few additional data; however we do not think this would significantly improve the interpretation of our data (as we focused on increase of MAP and brain physiology rather than describing the effects of a specific therapy of sepsis). Figures have been modified accordingly.
- A list of abbreviations would be helpful for the readers.
Authors’ response: This is not requested by the journal policy, but we would be more than happy to provide one if the Editor confirms this is necessary.
Round 2
Reviewer 2 Report
In the revised manuscript, entitled “Effects of reversal of hypotension on cerebral microcirculation and metabolism in experimental sepsis” by Taccone et al. the authors have improved the quality of the paper and several new paragraphs as well as 2 new references have been added. As most of important questions have been addressed in the revised manuscript, I have no further questions to raise, and now it is potentially acceptable for publication in Biomedicines.